# Lipopolysaccharide Core Truncation in Invasive *Escherichia coli* O157:H7 ATCC 43895 Impairs Flagella and Curli Biosynthesis and Reduces Cell Invasion Ability

**DOI:** 10.3390/ijms25179224

**Published:** 2024-08-25

**Authors:** Haiqing Sheng, Robinson J. Ndeddy Aka, Sarah Wu

**Affiliations:** Department of Chemical and Biological Engineering, University of Idaho, Moscow, ID 83844, USA; robinson@uidaho.edu

**Keywords:** invasive *E. coli* O157:H7, LPS core, production of flagella and curli, bovine host, cell invasion

## Abstract

*Escherichia coli* O157:H7 (*E. coli* O157) is known for causing severe foodborne illnesses such as hemorrhagic colitis and hemolytic uremic syndrome. Although *E. coli* O157 is typically regarded as an extracellular pathogen and a weak biofilm producer, some *E. coli* O157 strains, including a clinical strain ATCC 43895, exhibit a notable ability to invade bovine crypt cells and other epithelial cells, as well as to form robust biofilm. This invasive strain persists in the bovine host significantly longer than non-invasive strains. Various surface-associated factors, including lipopolysaccharides (LPS), flagella, and other adhesins, likely contribute to this enhanced invasiveness and biofilm formation. In this study, we constructed a series of LPS-core deletion mutations (*waaI*, *waaG*, *waaF*, and *waaC*) in *E. coli* O157 ATCC 43895, resulting in stepwise truncations of the LPS. This approach enabled us to investigate the effects on the biosynthesis of key surface factors, such as flagella and curli, and the ability of this invasive strain to invade host cells. We confirmed the LPS structure and found that all LPS-core mutants failed to form biofilms, highlighting the crucial role of core oligosaccharides in biofilm formation. Additionally, the LPS inner-core mutants Δ*waaF* and Δ*waaC* lost the ability to produce flagella and curli. Furthermore, these inner-core mutants exhibited a dramatic reduction in adherence to and invasion of epithelial cells (MAC-T), showing an approximately 100-fold decrease in cell invasion compared with the outer-core mutants (*waaI* and *waaG*) and the wild type. These findings underscore the critical role of LPS-core truncation in impairing flagella and curli biosynthesis, thereby reducing the invasion capability of *E. coli* O157 ATCC 43895.

## 1. Introduction

*Escherichia coli* O157:H7 (*E. coli* O157) is a Shiga-toxin-producing pathogen that poses significant public health risks due to its potential to cause severe foodborne illnesses, including hemorrhagic colitis and hemolytic uremic syndrome [1,2]. This bacterium colonizes the bovine recto-anal junction in healthy cattle, which act as primary reservoirs and are common source of foodborne infections [3,4]. The economic impact of *E. coli* O157 contamination in beef and fresh produce is considerable, with estimates suggesting that medical costs and lost productivity in the United States alone may reach USD 400 million annually [5]. Additionally, the economic burden on meat producers and vegetable growers can include product loss and negative publicity associated with product recalls [6]. Attachment to biological surfaces is a crucial first step in colonization, facilitated by various surface-associated factors such as lipopolysaccharides (LPS) [7], fimbriae [8], flagella, and other adhesins such as intimin [9,10,11], autotransporters [12], and curli [13]. These factors contribute to biofilm formation, cell adherence and invasion, and persistence in the bovine host.

LPS is an essential surface component of the outer membrane in Gram-negative bacteria. *E. coli* LPS is composed of three distinct regions: the O-antigen, the core oligosaccharide (OS), and lipid A [14]. In *E. coli* O157, the LPS core-OS is of the R3 type, which differs distinctly from that of *E. coli* K-12 [15]. The inner core of the R3 type comprises 3-deoxy-D-manno-oct-2-ulosonic acid (Kdo) and ADP-heptose residues, while the outer core is constructed of hexoses and 2-acetoamido-2-deoxy-hexoses. ADP-heptose is a crucial component of the LPS inner core, linking the outer part of LPS to Kdo between the Kdo2-lipid A and O-antigen. The R3 core structure involves several genes in the *waa* operon, including *waaC*, *waaF*, *waaQ*, *waaG*, *waaO*, *waaR*, *waaY*, *waaZ*, *rfaE*, and *rfaD* [15]. These genes collectively contribute to adding various sugar residues and modifications to the core oligosaccharide, resulting in a stable and functional LPS molecule. Mutations or deletions in these genes can lead to truncated LPS structures, which impact the overall functionality of the bacterial outer membrane [16,17]. The structural integrity of the outer membrane is crucial for the assembling and functionality of other surface structures such as flagella and curli, and for interactions with the host environment [14,18,19,20]. Disruptions in LPS-core synthesis can lead to significant alterations in bacterial behavior and pathogenic potential. Understanding the roles of these genes can provide valuable insights into how modifications in LPS structure can influence the virulence and survival strategies of *E. coli* O157. 

*E. coli* O157 is typically regarded as an extracellular pathogen and a weak biofilm producer [21,22]. However, some *E. coli* O157 strains, including a clinical strain ATCC 43895 (*E. coli* O157 43895), exhibit a notable ability to invade bovine crypt cells and other epithelial cells, as well as to form robust biofilm at 37 °C [23]. The invasive strain *E. coli* O157 ATCC persists significantly longer in the bovine host than non-invasive *E. coli* O157 strains [13]. The biofilm formation of *E. coli* O157 43895 is associated with LPS and curli. Its curli production promotes cell invasion [13]. Curli fimbriae, composed of polymerized amyloid protein, are produced by many *E. coli* and *Salmonella typhimurium* strains, preferentially at relatively lower temperatures (25–30 °C) [24,25]. We previously showed that *E. coli* O157 43895 cells produce curli at 37 °C [13]. The influence of LPS on surface structures such as flagella and curli, and consequently on invasion ability, is not yet fully understood. Our previous study showed that deletion of the genes involved in biosynthesis of *E. coli* O157 43895 O-antigen does not affect curli production [13]. However, other LPS biosynthesis genes, such as *waaG*, *rfbH*, and *lpxM* have been implicated as important for curli production [26,27,28]. 

In this study, we constructed derivative strains of *E. coli* O157 43895 with a series of ordered LPS-core deletion mutations, leading to stepwise truncations of the LPS. We investigated how truncating the LPS core-OS affects the behaviors of invasive *E. coli* O157 43895, particularly its interaction with epithelial cells. By deleting specific genes involved in LPS core-OS synthesis and comparing these strains with the wild type and previously created O-antigen mutants, we aimed to understand the impacts on key virulence factors, such as flagella and curli biosynthesis, as well as the bacterium’s ability to invade host cells. The findings offer valuable insights into the molecular mechanisms underlying *E. coli* O157 persistence and pathogenicity, potentially identifying targets for interventions to reduce bovine carriage and human infections caused by this pathogen.

## 2. Results

Construction of *E. coli* O157 43895 LPS mutants and comparison of their LPS structures. We generated a series of LPS mutants of invasive *E. coli* O157 43895, each exhibiting progressively shorter LPS structures, as illustrated in Figure 1. Four genes involved in core biosynthesis were individually deleted from the chromosome of *E. coli* O157 43895, resulting in the mutants Δ*waaI*, Δ*waaG*, Δ*waaF*, and Δ*waaC*. Additionally, three O-antigen mutants (*wzy*-Tn5, *per*-Tn5, and *manC*-Tn5) generated previously by Tn5 mutagenesis were included in this study for their LPS structural analysis and comparison [13]. The expected structures of LPS in these seven *E. coli* O157 43895 mutants are shown in Figure 1A. 

The mutations commenced with a *wzy* mutant, characterized by LPS containing only a single O-antigen unit, and culminated in a *waaC* mutant with an inner core truncated to the Kdo residues (Figure 1A). Each mutation yielded a strain with an LPS length consistent with our expectations. The wild-type strain produced complete LPS molecules with long O-antigens, while the LPS of the *waaC* mutant comprised only lipid A and Kdo. The LPS synthesized by the other mutants exhibited varying degrees of core and/or O-antigen sugar presence, as visualized in the gel (Figure 1B).

Comparison of biofilm formation of *E. coli* O157 43895 LPS and flagella mutants. We previously demonstrated that the invasive *E. coli* O157 43895 forms robust biofilm, even at an elevated temperature of 37 °C, compared with non-invasive strains [13]. In this study, we compared the biofilm-formation abilities of O-antigen, core-OS, and flagella mutants of *E. coli* O157 43895 with the WT by observing pellicle biofilms and conducting quantification measurement. The WT, along with the O-antigen mutants *wzy*-Tn*5*, *per*-Tn5, and *manC*-Tn5, the LPS-core mutants Δ*waaI*, Δ*waaG*, Δ*waaF*, and Δ*waaC*, and the flagella mutant Δ*fliC*, were grown in LB broth overnight at 37 °C under static condition. After 48 h of culture, pellicles and rings were visible in the WT and the flagella mutant (Figure 2A). The O-antigen mutants (*wzy*-Tn*5, per*-Tn5, and *manC*-Tn5) also formed visible rings, though they were less pronounced compared with the WT. In contrast, truncation of the LPS core abolished pellicle biofilm formation, as no rings were observed in the cultures of the core mutants (Δ*waaI*, Δ*waaG*, Δ*waaF*, and Δ*waaC*) (Figure 2A). 

In the quantification assay, the WT and mutant strains were grown in a 96-well microtiter plate for 24 h. Biofilms were stained with crystal violet and measured using a spectrophotometer. The Δ*fliC* mutant demonstrated a biofilm formation ability with an OD595 reading of 0.46, comparable to the WT’s reading of 0.54 (*p* > 0.05), suggesting that biofilm formation of *E. coli* O157 43895 is not dependent on flagella. In contrast, biofilm formation by O-antigen mutants (*wzy*-Tn*5, per*-Tn5, and *manC*-Tn5) was significantly reduced compared with the WT (*p* ≤ 0.01), with the OD595 reading ranging from 0.25 to 0.31 (Figure 2B). Biofilm formation was even further diminished in the LPS-core mutants (Δ*waaI*, Δ*waaG*, Δ*waaF*, and Δ*waaC*), with the OD595 reading in the 0.1 range (*p* ≤ 0.001). The reduction was consistent with the absence of a pellicle ring in the tube assays, indicating that these LPS-core mutants lost their ability to form biofilm. The significant difference in biofilm formation between the LPS core-OS mutants and the O-antigen mutants highlights the importance of an intact of LPS core for biofilm formation in *E. coli* O157 43895.

Cell growth of *E. coli* O157 43895 WT and LPS-core mutants. To investigate the impact of the LPS core on cell growth, core mutants and the WT were grown in MOPS media at 37 °C for 24 h with aeration. As shown in Figure 3, the two outer-core mutants, Δ*waaI* and Δ*waaG*, displayed growth comparable to the WT strain throughout the 24 h period. In contrast, the two LPS inner-core mutants, Δ*waaF* and Δ*waaC*, experienced an expanded lag phase and grew significantly slower than the WT during the logarithmic phase (4 h to 12 h) (*p <* 0.001); however, the two mutants were able to eventually catch up in growth rate during the stationary phase. These results suggest that the absence of the inner LPS core initially affects cell growth.

Motility, presence of flagella and curli. We previously demonstrated that the O-antigen mutant Δ*per* of *E. coli* O157 43895 has impaired motility but can be induced to swim after extended incubation in 0.3% soft agar [7]. Here, we determined the relative ability of LPS-core mutants of *E. coli* O157 43895 to swim on soft agar and the presence of flagella. After 48 h of prolonged incubation, the O-antigen mutants and one of the outer-core LPS mutants Δ*waaI* were able to swim to varying degrees on soft agar, though with impaired motility. In contrast, three core LPS mutants, Δ*waaG*, Δ*waaF*, and Δ*waaC*, lost motility (Figure 4). Western-blot analysis indicated that these three mutants had defective motility due to lack of flagella (Figure 4). 

Unlike non-invasive *E. coli* O157 strains, ATCC 43895 produces curli at 37 °C. To investigate the influence of the core-OS on curli production, we grew the LPS mutants of the invasive strain on CRI plates at 37 °C. After 24 h incubation, the WT and the O-antigen and outer-core mutants (Δ*waaI and* Δ*waaG*), and flagella mutant (Δ*fliC*) all produced curli, as evidenced by red coloration of the colonies due to Congo-red binding on the CRI plate. In contrast, the two inner-core LPS mutants (Δ*waaF and* Δ*waaC*) remained white, indicating a lack of curli production (Figure 5). 

The effect of LPS core-OS of O157 43895 on cell adherence and invasion. *E. coli* O157 43895 has a strong ability to adhere to and invade epithelial cells compared with other *E. coli* O157 strains. We previously demonstrated that truncation of O-antigen in *E. coli* O157 43895 (*wzy*-Tn*5*, *per*-Tn5, and *manC*-Tn5) did not affect its cell adherence and invasion [13]. In this study, we examined the invasion ability of the isogenic LPS-core mutants Δ*waaI*, Δ*waaG*, Δ*waaF*, and Δ*waaC*, as well as the flagella mutant Δ*fliC*. The WT and mutants were co-cultured with MAC-T cells individually. Giemsa staining revealed that the outer-core mutants Δ*waaI* and Δ*waaG*, and mutant Δ*fliC*, exhibited a large number of bacteria adhering to the epithelial cells in an aggregative pattern, similar to WT. However, the two inner-core mutants, Δ*waaF* and Δ*waaC*, showed dramatically reduced adherence to MAC-T cells, with only a few sporadic bacteria adhering to the cells (Figure 6A). Subsequently, we assessed the ability of these mutants to invade epithelial cells. The WT and mutants were individually co-cultured with a monolayer of MAC-T cells in a 24-well plate at MOI of 10 for 3 h. The number of bacteria recovered after 2 h of gentamicin treatment indicated the number of intracellular bacteria. Among the four LPS-core mutants, Δ*waaF* and Δ*waaC* showed a more than 100-fold and 10-fold reduction in cell invasion, respectively, compared with WT *E. coli* O157 43895 and curli-deficient *E. coli* O157:H7 ATCC 43894. The log value of intracellular bacteria for the two inner-core mutants was significantly lower compared with the WT and the outer-core mutants Δ*waaI* and Δ*waaG* (*p* < 0.001), as well as to the non-invasive strain ATCC 43894 (*p* < 0.01). 

## 3. Discussion

The LPS core of *E. coli* O157 belongs to the R3 type, which is preferentially associated with key virulence determinants such as Shiga toxins [29,30,31] and the adhesin intimin and its receptor Tir [32,33]. In this study, we systematically investigated the relationship between O-antigen/LPS-core length and biofilm formation, flagella and curli production, and cell adherence and invasion. Deletion of the *waa* or disruption of *wba* genes targeted in this study resulted in truncated LPS structures (Figure 1). The results demonstrate the critical role of the LPS core in various cellular functions of *E. coli* O157 43895.

Unlike non-invasive *E. coli* O157 strains, ATCC 43895 forms a robust pellicle biofilm at 37 °C. Pellicle biofilms are bacterial communities at the air–liquid interface, composed of extracellular polymeric substances such as polysaccharides, proteins, flagella, fimbriae, and nucleic acids secreted by the bacteria [34,35]. In some Gram-negative bacteria, LPS is crucial for biofilm matrix construction [36,37,38]. This extracellular matrix can enhance bacterial adherence to host tissues, facilitating colonization in bovine hosts and potentially leading to deeper tissue invasion [23]. Previous research has shown that disrupting the O-antigen biosynthetic genes (*per*, *manC*, and *wzy*) in *E. coli* O157 43895 and curli fibers contribute to biofilm formation [13]. In this study, we compared the biofilm formation abilities of O-antigen, LPS-core, and flagella mutants of *E. coli* O157 43895 with the wild type. We found that truncating the LPS core completely abolished biofilm formation, highlighting the critical role of a complete LPS core in maintaining biofilm integrity. The differences in biofilm formation among LPS-core and O-antigen mutants suggest additional factors beyond LPS-core length. Interestingly, despite lacking motility, the flagella *fliC* mutant still formed a robust pellicle biofilm, which was unexpected.

We analyzed the mobility and flagella presence in O-antigen and LPS-core mutants of invasive *E. coli* O157 43895. All mutants exhibited significantly reduced motility, with Δ*waaG*, Δ*waaF*, and Δ*waaC* mutants completely losing motility. Western-blot analysis revealed that these mutants were unable to produce flagella. Similar findings were observed in *E. coli* K-12; LPS-core mutants showed no flagella on their cell surfaces under electron microscopy compared with the wild type [39]. Non-motile phenotypes in LPS-deficient strains have also been reported in other Gram-negative bacteria. For instance, deep rough mutants Δ*rfaG* and Δ*rfaD* in Salmonella, which lack outer- and inner-core components, respectively, showed dramatically impaired motility [40,41]. These results suggest that truncation of the LPS core affects flagella assembly in both *E. coli* and *Salmonella*. In contrast, truncated LPS-core mutants of *Pseudomonas aeruginosa* PAO1, such as those with *rmlC*, *migA*, and *wapR* deletions, exhibited reduced motility, but flagella assembly appeared intact [42]. 

We also assessed curli production in the O-antigen and LPS-core mutants using a Congo-red-binding assay. Compared with O-antigen and outer-core mutants, as well as the wild type, the Δ*waaF* and Δ*waaC* mutants, which are defective in the inner core, failed to produce curli on their surfaces. This finding is consistent with an earlier study that screened the Keio collection of single-gene deletions in *E. coli* K-12 using Congo-red indicator plates and identified LPS-core mutants of *E. coli* K-12 that were defective in curli production [43]. These results support the notion that LPS integrity is crucial for maintaining outer membrane stability and the proper assembly of cell-surface components. Truncation of the LPS core disrupts the outer membrane, leading to leakage of periplasmic contents into the extracellular space and alterations in outer membrane composition [44].

It remains unclear whether the altered LPS structure or the mutated genes involved in LPS-core biosynthesis directly caused the defects in flagella and curli fimbriae. The truncation of the LPS core might induce general cell-envelope stress, downregulating flagella and curli assembly in mutants like Δ*waaF* and Δ*waaC*, which could affect interactions with epithelial cells. Future studies will focus on these aspects.

We observed that Δ*waaF* and Δ*waaC* mutants, which lack heptose, experienced an extended lag phage at 37 °C, suggesting that heptose incorporation into the inner core is crucial for optimal cell growth, particularly during the early stages. Interestingly, Murata et al. reported that certain LPS synthetic genes involved in heptose biosynthesis (*gmhA*, *gmhB*, and *gmhD*) or its incorporation into the LPS inner core (*waaC* and *waaF*) are essential for bacterial growth at critical high temperatures in *E. coli* K-12 [45].

Lastly, the reduced adherence and invasion abilities of inner-core mutants Δ*waaF* and Δ*waaC* underline the necessity of a complete LPS core for effective interaction with host cells. These findings suggest that the LPS-core structure not only contributes to production of flagella and curli and supports growth but also plays a crucial role in pathogenesis by mediating adherence and invasion in invasive *E. coli* O157 43895.

In summary, this study underscores the multifaceted role of the LPS core in invasive *E. coli* O157 43895, affecting growth, motility, biofilm formation, and host cell interactions, with significant implications for understanding bacterial persistence and developing targeted interventions. 

## 4. Materials and Methods

Bacterial strains, plasmids, media, and growth conditions. The bacterial strains, plasmids, and primers used in this study are listed in Table 1 and Table 2. Bacteria were grown in Luria–Bertani (LB) broth or agar at 37 °C, unless otherwise stated. When required, antibiotics (Sigma-Aldrich, St. Louis, MO, USA) kanamycin (Kan, 50 mg/mL), chloramphenicol (Cm, 30 mg/mL), or ampicillin (Amp, 100 mg/mL) were added to the media. Congo-red-dye-binding of curli was monitored on Congo-red-indicator (CRI) agar: 10 g Casamino acids, 1 g yeast extract, 20 g agar, 20 mg Congo red, and 10 mg Coomassie brilliant blue G-250/L (Sigma-Aldrich).

Mutant-strain construction. To create a mutant deficient in expression of LPS core-OS in *E. coli* O157 43895, the Lambda Red recombinase system [46] was used for gene deletion, as previously described (7). Oligonucleotide primers were purchased from Invitrogen (Carlsbad, CA, USA). The genes of the LPS core-OS of *E. coli* O157 43895 were independently replaced by a chloramphenicol resistance cassette (the *cat* gene from pKD3). The genes *waaI*, *waaG*, *waaF*, and *waaC* were selected for deletion based on their position in the *waa* cluster. As shown in Figure 1A, *waaC* encodes a heptosyltransferase I enzyme, which adds the first L-glycero-D-manno-heptose (Hep) residue to the Kdo (3-deoxy-D-manno-oct-2-ulosonic acid) molecule in the LPS core-OS; *waaF* encodes a heptosyltransferase II enzyme, responsible for the addition of the second Hep residue to the growing LPS core-OS oligosaccharide; both *waaG* and *waaI* encode a glucosyltransferase that adds a glucose (Glc) residue to the LPS core-OS (Figure 1A). In addition, H7 flagella mutant was made by deleting *fliC* for comparison. PCR primers for deletions of *waaI*, *waaG*, *waaF*, and *waaC* were designed based on the sequence of the *waa* cluster in the *E. coli* O157 43895 genome (accession number NZ_CP008957). Chloramphenicol-resistant (Cm^R^) recombinants were selected on LB agar plates containing chloramphenicol. The O-antigen mutants *manC*-Tn*5, per*-Tn5, and *manC*-Tn5*,* generated previously, were included in this study for comparison [13]. The failure of the LPS core-OS mutants to express the O157 antigen was confirmed by anti-O157 latex agglutination (Pro-Lab Diagnostics, Toronto, ON, Canada). The chloramphenicol resistance gene was eliminated. The replacement of each of the targeted genes by a nonpolar scar structure [46] and the mutations were verified by PCR.

LPS analysis. To confirm the loss of LPS core-OS in the mutants, LPS was isolated and analyzed using a previously described method [47]. Briefly, cells were grown in 3 mL LB broth to an optical density at 600 nm (OD600) of 1.0. The cells were then harvested by centrifugation, resuspended in 1.0 mL of phosphate-buffered saline, and incubated at 60 °C for 30 min. The suspension was then centrifuged at 12,000× *g* for 30 min, and the supernatant was mixed with an equal volume of Tricine sample buffer (Bio-Rad Laboratories, Hercules, CA, USA) and boiled for 10 min. Proteinase K was added to a final concentration of 0.5 mg/mL, and the sample was incubated at 60 °C for 60 min before being centrifuged at 16,000× *g* for 30 min. LPS was then analyzed by Tricine–SDS-PAGE and visualized by silver staining, as previously described [7].

Crystal violet (CV) biofilm assay. To assess the ability of *E. coli* O157 mutants to produce a pellicle biofilm at the air–liquid interface, a fresh colony was grown in LB broth overnight at 37 °C. This culture was then diluted 500-fold with fresh 4 mL LB in a 15 mL polystyrene tube and incubated overnight at 37 °C under static conditions. The content of each tube was removed. The tubes were stained with 5 mL of 1% crystal violet solution for 5 min. The dye was then removed, and the tubes were rinsed with water and air dried. 

Biofilm quantification assays were performed using a standard microtiter assay [48]. The overnight culture was inoculated into duplicate 96-well microtiter plates containing minimal salt medium (MSM) with 0.04% glucose and incubated for 24 h without agitation at 37 °C. Each isolate was inoculated in triplicate. After incubation, the titer plates were washed with water and stained with 1% crystal violet for 15 min at room temperature. Plates were then rinsed vigorously with water again to remove unattached cells and residual dye. Biofilm formation was evaluated by measuring the absorbance of the solubilized dye in 95% ethanol at a wavelength of 595 nm using a PowerWave XS reader (Bio-Tek, Winooski, VT, USA). 

Growth curves in MOPS media. Three isolates of each strain were cultured overnight in LB broth at 37 °C with shaking at 150 rpm. Overnight cultures were diluted with LB to 0.1 OD600 and grown for 4 h at 37 °C. Cells were pelleted then washed twice with 1× PBS and resuspended in MOPS media [49] (1× MOPS supplemented with 0.5% glucose, 1 µg/mL thiamine, 0.1 mM K2HPO4). The cells in MOPS were diluted to approximately 0.1 OD600. Two hundred microliters of cells was pipetted in technical replicates of 3 in a microtiter plate and growth was monitored using the PowerWave XS reader at 37 °C with continuous shaking. Optical density measurements were collected at a wavelength of 600 nm every 30 min, 5 s after shaking stopped for 24 h.

Motility assays. The swimming motility of the wild-type and the mutant strains was assessed using tryptone swarm plates containing 1% Bacto Tryptone, 0.5% NaCl, and 0.3% agar. Five microliters of an overnight culture was point-inoculated into swarm plates and incubated at 37 °C for up to 40 h. The diameter of the motility halo was measured. 

Adherence and invasion assays. The bacterial internalization of MAC-T cells was measured using a standard gentamicin protection assay. First, 24-well tissue culture plates (Corning Costar, NY, USA) were seeded with 10^4^ MAC-T cells and incubated in 5% CO_2_ at 37 °C until cells were confluent. The MAC-T cell monolayers were then washed twice with Hank’s buffered saline solution (HBSS), and approximately 2 × 10^6^ *E. coli* O157 cells (multiplicity of infection, MOI, 10:1) in cell culture medium without antibiotics or fetal bovine serum (1 mL) were added. After 3 h at 37 °C, unattached extracellular bacteria were removed by suction, and the monolayers were washed three times with HBSS. Fresh medium containing 100 µg/mL gentamicin was added to kill extracellular bacteria. After an additional 2 h incubation, the monolayers were washed three times with HBSS without Ca^2+^ or Mg^2+^. Epithelial cells were lysed by adding 100 µL 0.5% trypsin-EDTA and 900 µL 0.05% Triton X-100/well for 5 min. Bacterial invasion was quantified by counting the CFUs recovered/well on LB agar. To microscopically view the interaction of the bacteria with MAC-T cells, the cells on cover slips were fixed with methanol, stained with Giemsa stain (0.4% *w*/*v*, Sigma-Aldrich), and examined microscopically under oil immersion at 100X.

Flagella isolation and immunoblotting. Flagella were sheared from *E. coli* O157 derivatives using a standard protocol [50] with slight modification. Briefly, *E. coli* O157 strains were cultured on motility agar for 40 h, as mentioned above. LB was inoculated with 20 µL agar plugs from the edge of the motility halo and incubated at 37 °C with shaking for 30 h. Bacterial pellets were collected by centrifugation (5000× *g*) for 15 min at 4° C, and resuspended with 1 mL cold PBS with protease inhibitor (Sigma-Aldrich) at appropriate concentration. The microtubes with the bacterial suspension were put on ice for 15 min, and placed a FastPrep FB120 Cell Disruptor (Abiogene, CA, USA) at a 5.5 m/sec speed for 30 s. The vigorous vortex step was repeated 4 times and the tubes were put on ice for 10 min at each interval. The supernatants were transferred into clean tubes after centrifuge (8000× *g*) for 15 min at 4 °C. The cell fibers were pelleted by centrifugation (41,000× *g* for 3 h at 4 °C). To verify the presence of flagella, sheared fibers were monomerized according to a published protocol [51]. Flagella preparations were incubated at 70 °C for 15 min and then were examined by Western blotting with the 15D8 monoclonal antibody specific for enteric flagella [52]. Cell suspensions from overnight cultures grown in LB were extracted by boiling, fractionated by SDS-PAGE gel electrophoresis, transferred onto PVDF membranes using a Mini Trans-Blot electrophoretic transfer cell (Bio-Rad, Redmond, WA, USA), and probed with 15D8 (1:2000).

Statistical analysis. Statistical analyses were performed by using GraphPad Prism software 7 (San Diego, CA, USA). Details of sample size, test used, error bars and statistically significant cutoff are presented in the text or figure legends. 

## Figures and Tables

**Figure 1 ijms-25-09224-f001:**
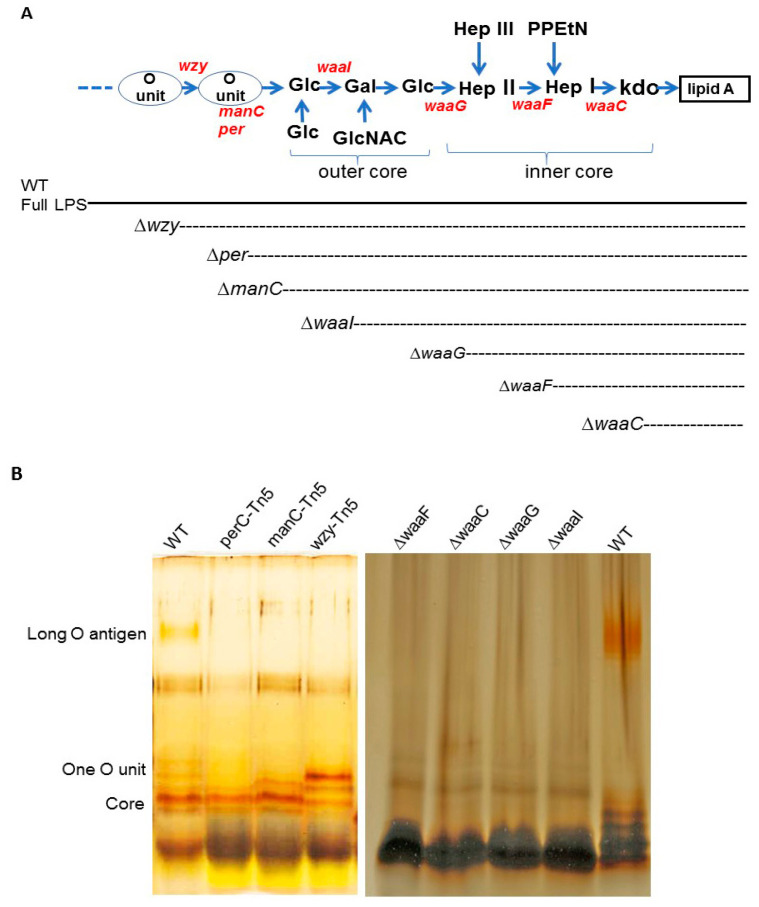
Structure and analysis of *E. coli* O157 LPS molecules. (**A**) Schematic representation of *E. coli* O157 LPS molecule, showing O-antigen, outer core, inner core, and lipid A. Dotted lines indicate the level of LPS truncation resulting from each mutation (Kdo, 3-deoxy-D-manno-octulosonic acid; PPEtN, pyrophosphorylethanolamine; Hep, heptose; GlcNAc, N-acetylglucosamine; Glc, glucose; Gal, galactose). (**B**) LPS was isolated from wild-type and mutant cells of *E. coli* O157 43895 and analyzed by silver-stained Tricine–SDS-PAGE, as described in the Materials and Methods section. The expected location of O-antigen regions and the core is indicated on the left side of the gel.

**Figure 2 ijms-25-09224-f002:**
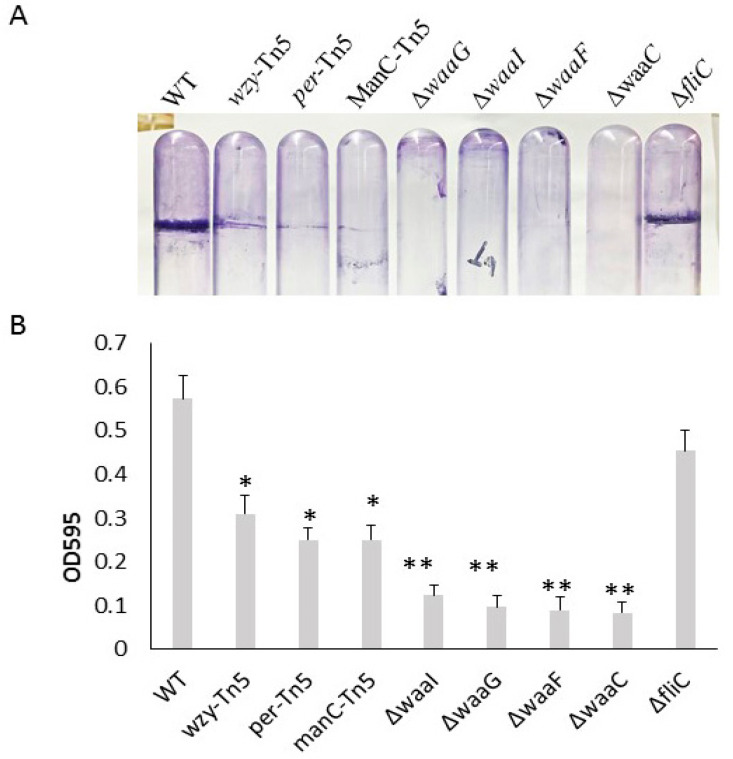
Biofilm formation of *E. coli* O157 43895 WT and mutant derivatives. (**A**) Pellicle formation at 37 °C. Pellicle rings of a 48 h old floating biofilm of *E. coli* O157 43895. Mutants were detected by crystal violet (CV) staining. (**B**) Biofilm measurement using a standard microtiter assay. *E. coli* O157 43895 WT and mutants were incubated at 37 °C in MSM with 4% glucose. Bound CV was solubilized with 95% ethanol and quantified by absorbance at 595 nm. Assays were performed six times for each strain. Mean values with standard deviation are shown. Statistical analysis used one-way ANOVA test and Dunnett post-test. * *p* < 0.01, ** *p* < 0.001, compared with the wild type (WT).

**Figure 3 ijms-25-09224-f003:**
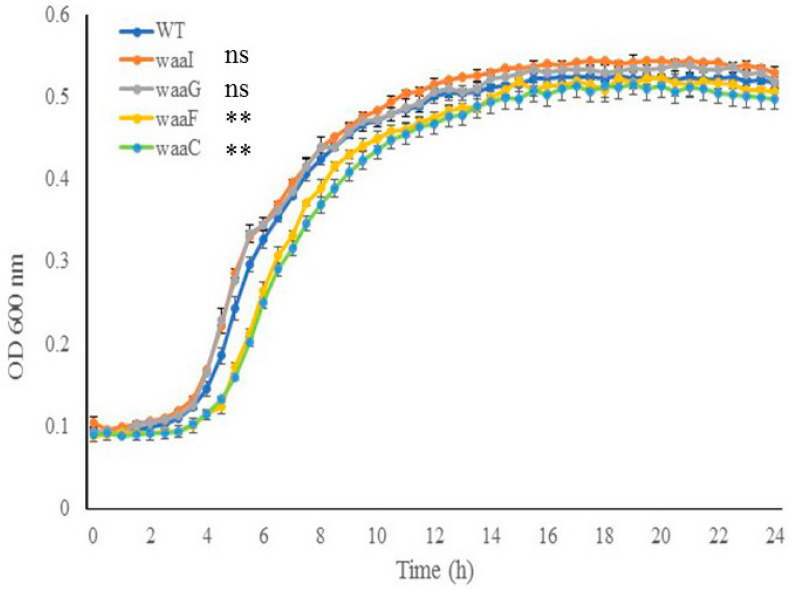
The effect of LPS-core truncation on cell growth. Growth of wild-type and LPS mutant cells (Δ*waaI*, Δ*waaG*, Δ*waaF*, and Δ*waa*C) in MOPS media was monitored using the PowerWave XS plate reader (37 °C with shaking, wavelength 600 nm). Error bars represent standard deviation of the mean for 3 replicates. Error bars not shown for data points where standard deviation is smaller than symbol dot itself. The data of the mutants during log phase (4 h to 12 h) were compared with that of the wild type (WT). ** *p <* 0.001, ns., nonsignificant via two-way ANOVA with Tukey’s test.

**Figure 4 ijms-25-09224-f004:**
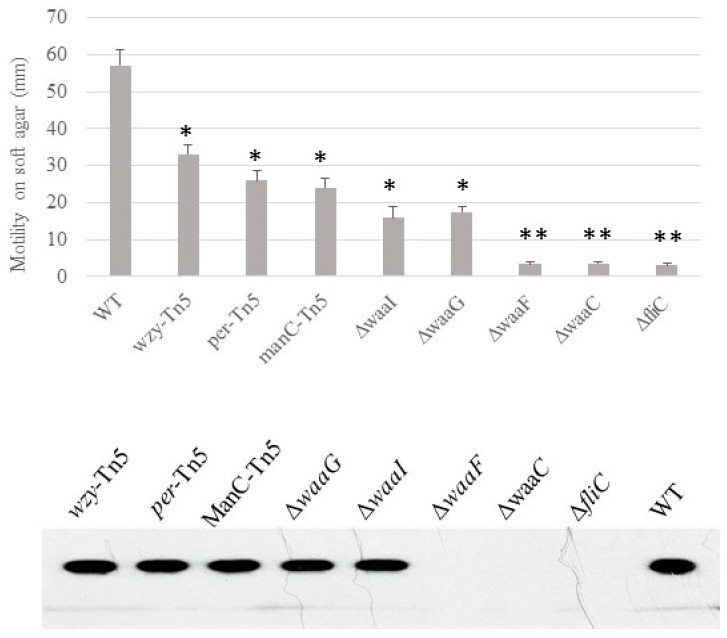
The effect of LPS-core truncation on the swimming motility and flagella biosynthesis of *E. coli* O157 43895. Swimming motility of wild-type and LPS mutants was assessed using soft-agar swarm plates. After 40 h incubation, the diameter of the motility halo was measured. Experiments were repeated three times. Presence of flagella was detected by immunoblot (lower panel). Mean values with standard deviation are shown. Statistical analysis used two-tailed unpaired *t*-test. * *p* < 0.01, ** *p* < 0.001, compared with the wild type (WT).

**Figure 5 ijms-25-09224-f005:**
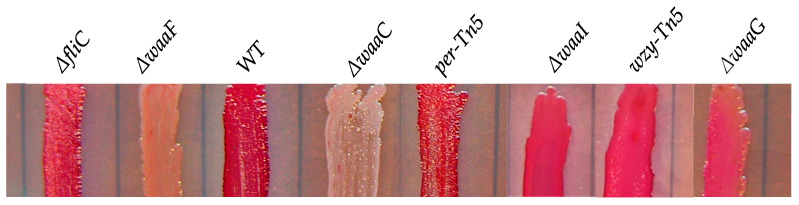
The effect of LPS-core truncation on the curli biosynthesis of *E. coli* O157 43895. *E. coli* O157 43895 and mutants were grown on Congo-red (CR) indicator plates at 37 °C for 24 h. Presence of curli on the cell surface was detected by CR binding.

**Figure 6 ijms-25-09224-f006:**
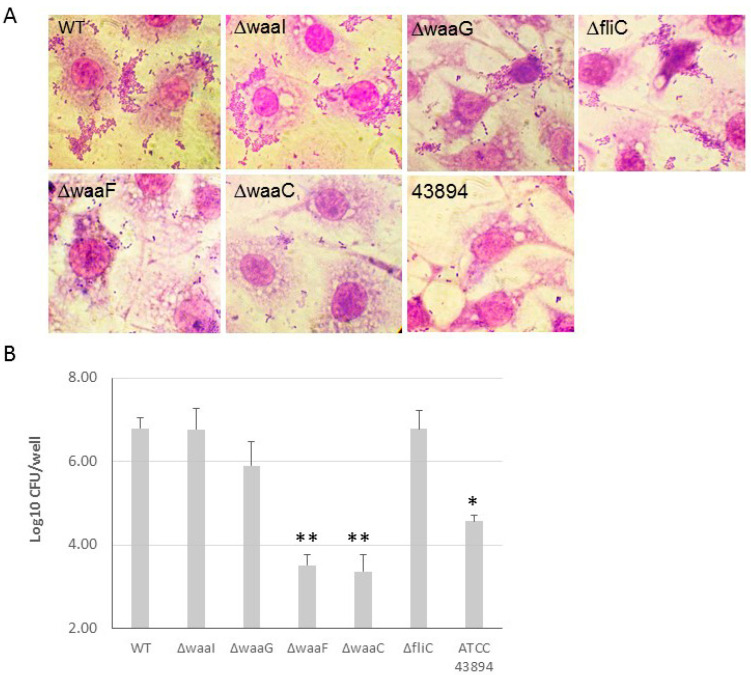
The effect of LPS core-OS truncation on cell adherence and invasion. (**A**) The adherence of LPS-core and flagella mutants of *E. coli* O157 43895 to epithelial cells was examined. The monolayers of MAC-T cells on coverslips in 24-well plates were co-cultured with the bacteria at 37 °C for 3 h. The cells were washed five times with PBS and fixed with 4% paraformaldehyde for 15 min, washed again with PBS, stained with Giemsa stain, and viewed under the microscope (magnification ×1000). (**B**) *E. coli* O157 43895 and the mutants were tested for epithelial-cell invasion using a MAC-T gentamicin protection assay, as described in the Materials and Methods section. Bacteria were co-cultured with MAC-T monolayers at a multiplicity of infection (MOI) of 10. The experiment was repeated three times with three replicates per strain, and CFUs were determined by plate count and were transformed to log10 value. Mean values with standard deviation are shown. Statistical analysis used one-way ANOVA test and Dunnett post-test. * *p* < 0.01, ** *p* < 0.001, compared with the wild type (WT).

**Table 1 ijms-25-09224-t001:** *E. coli* O157 strains and plasmids.

Strain or Plasmid	Description	Source or Reference
43895	*E. coli* O157:H7 ATCC 43895, a clinical isolate, stx*1*+/*stx2*+, curli-positive	ATCC ***
*manC*-Tn*5*	Tn*5* inserted in *manC* of 43895	[13]
*per*-Tn*5*	Tn*5* inserted in *per* of 43895	[13]
*wzy*-Tn*5*	Tn*5* inserted into *wzy* of 43895	[13]
*∆* *waaI*	43895 with *waaI* deletion	This work
*∆* *waaG*	43895 with *waaG* deletion	This work
*∆* *waaF*	43895 with *waaF* deletion	This work
*∆* *waaC*	43895 with *waaC* deletion	This work
*∆fliC*	43895 with *fliC* deletion	This work
pKD3	Template plasmid for mutagenesis (Amp^R^,Cm^R^)	[46]
pKD46	Red recombinase helper plasmid, RepA101(Ts), Amp^R^	[46]
pCP20	Temperature-sensitive replicon, thermal induction of FLP synthesis, Amp^R^,Cm^R^	[46]

*** ATCC, American Type Culture Collection, Manassas, VA.

**Table 2 ijms-25-09224-t002:** Primers used in this study.

Primer Name	Sequence (5′-3′)
waaI-F	ttagaagcatttttctttataatactttaaataattaaaaatGTGTAGGCTGGAGCTGCTTCG
waaI-R	atgtctcaactcaatgatagtgacatcatcctttttgagtataCATATGAATATCCTCCTTA
waaG-F	tcaaccatccagaccacccgttatgatatccgccgctttctctGTGTAGGCTGGAGCTGCTTCG
waaG-R	atgatcgttgctttttgtttatataaatattttccctttggcggtCATATGAATATCCTCCTTA
waaF-F	atgaaaatactggtgatcggcccgtcttgggttggcgacatgaGTGTAGGCTGGAGCTGCTTCG
waaF-R	tcaggcttcctcttgtaacaatagcgcgttgagttctttcagtaCATATGAATATCCTCCTTA
waaC-F	atgcgggttttgatcgttaaaacatcgtcgatgggcgatgttctGTGTAGGCTGGAGCTGCTTCG
waaC-R	tcatcttatctccgatgtcaacttattggttattatttcaagaCATATGAATATCCTCCTTA
fliC-F	ttaaccctgcagcagagacagaacctgctgcggtacctggttaGTGTAGGCTGGAGCTGCTTCG
fliC-R	atggcacaagtcattaataccaacagcctctcgctgatcactcaCATATGAATATCCTCCTTA

Homology extensions of targeted genes are indicated in lowercase letters.

## Data Availability

The data that support the findings of this study are available from the corresponding authors upon request.

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
