# Peer review of "Lipopolysaccharide Core Truncation in Invasive Escherichia coli O157:H7 ATCC 43895 Impairs Flagella and Curli Biosynthesis and Reduces Cell Invasion Ability"

_ijms, 2024, doi:10.3390/ijms25179224_

Round 1

Reviewer 1 Report

Comments and Suggestions for Authors

Comments to the Authors of manuscript number ijms-3169348 entitled “Lipopolysaccharide Core Truncation in Invasive Escherichia coli O157:H7 ATCC 43895 Impairs Flagella and Curli Biosynthesis and Reduces Cell Invasion Ability

Escherichia coli O157

(E. coli O157) is usually noninvasive and produces weak biofilms. However, some strains form strong biofilms at 37 °C and invade bovine and other epithelial cells. The invasive strain E. coli O157 ATCC 43895 persists longer in bovine hosts compared to non-biofilm-producing strains. This study explores how mutations in LPS-core genes (waaI, waaG, waaF, waaC) affect biofilm formation, flagella and curli production, and host cell invasion.

1. The introduction is quite lengthy and somewhat repetitive, making it hard to follow. For example, lines 39-43 and 44-48 both discuss the structure and role of LPS core-OS but could be streamlined.

2. the detailed description of the LPS structure (lines 40-49) might be condensed to focus more on how these structures relate to the research question.

3. lines 19-20 mention that E. coli O157 is a weak biofilm producer but do not directly connect this fact to the study's focus on specific strains and LPS mutations.

4. The use of technical jargon and abbreviations (e.g., LPS, core-OS, curli) could be minimized or explained

5. lines 83-85 and 126-127 describe LPS mutants and their effects on biofilm formation in depth, but the connection to the study's broader goals is not immediately clear.

6. lines 131-133 mention significant differences but do not provide numerical data or specific statistical values.

7. Discussion: how do these results compare with findings from other studies on LPS mutants?

8. Discussion:There is some repetition of concepts, particularly regarding the role of the LPS core in biofilm formation and motility.

9. Only the Student’s t-test is mentioned, but it is not specified in which situations exactly it was used (e.g., for comparisons of two groups, for parametric data, or for paired tests). There are different variants of the t-test (e.g., independent samples t-test, dependent samples t-test), and the lack of precise indication of which one was used can lead to ambiguity.

10. In studies comparing more than two groups or different conditions, analyses such as ANOVA (analysis of variance) are often used. If such analyses were performed, they should be mentioned. The absence of this information may suggest that potential differences between more than two groups were not assessed or that the analysis was limited.

11. Student’s t-test requires certain assumptions to be met, such as normality of the data distribution and homogeneity of variance. If the data do not meet these assumptions, other tests or statistical methods may be needed. It should be mentioned whether these assumptions were checked and how this affected the choice of statistical methods.

12. It should be reported what level of significance (e.g. p < 0.05) was used to judge the statistical significance of the results. Without this detail, it is not clear what criteria were used to consider the results statistically significant.

13. Information about the number of replicates and experiments that were performed is crucial to assessing the credibility and reliability of the results. Just stating in general that a test was performed does not provide information about how the accuracy and reliability of the results were ensured.

Reviewer 2 Report

Comments and Suggestions for Authors

The manuscript titled “Lipopolysaccharide Core Truncation in Invasive Escherichia coli O157:H7 ATCC 43895 Impairs Flagella and Curli Biosynthesis and Reduces Cell Invasion Ability” by Sheng, H.; et al. is a scientific work where the authors designed lipopolysacharide (LPS) mutant constructs to evaluate how the implemented mutations can affect the bacterial flagella and curli biogenesis and the adhesive film formation properties. The bacterial strain chosen in this research is involved in the attack of bovine individuals affecting to the cattle Industry and the agronomy sector. For this reason, this is a topic of growing interest. However, it exists some points that need to be addressed (please, see them below detailed point-by-point) to improve the scientific quality of the submitted manuscript paper before this article will be consider for its publication in the International Journal of Molecular Sciences.

1) The authors should consider to add the term “bovine host” in the keyword list.

2) “Escherichia coli O157:H7 (E.coli O157) is a pathogenic strain (…) foodborne illnesses, including hemorrhagic colitis and hemolytic uremic syndrome (…) persistence in bovine host” (lines 32-39). Could the authors provide quantitative data insights about the worldwide economic burdens according to the cattle sector? This will significantly aid the potential readers to better understand the significance of this devoted research.

3) “LPS are crucial surface components (…) membrane in Gram-negative bacteria E. coli (…) essential components that bridges the lipid A anchor to O-antigen polysaccharide (…) host environment” (lines 40-44). Here, even if I agree with this statement provided by the authors it is necessary to remark how the local nanomechanical cues [1] are affected by the incorporation of LPS inner the solid-phase membrane regions increasing not only the cellular membrane roughness, but also the mechanical performance [2]. This evidences that LPS can act as pivotal actor in temperature-sensitive lipid formation in the bacterial outer membrane enhancing the molecular cell surface recognition processes.

[1] Magazzù, A.; et al. Investigation of Soft Matter Nanomechanics by Atomic Force Microscopy and Optical Tweezers: A Comprehensive Review. Nanomaterials 2023, 13, 963. https://doi.org/10.3390/nano13060963.

[2] Kiss, B.; et al. Development, structure and mechanics of a synthetic E. coli outer membrane model. Nanoscale Adv. 2020, 3, 755-766. https://doi.org/10.1039/d0na00977f.

4) “Motility, presence of flagella and curli” (lines 155-186). Could the authors provide some microscopy images (e.g. by scanning electron microscopy o other complementary technique) to better visualize the morphology and state of the flagella and curli among the different examined bacterial specimens discussed in this section?

5) Figure 6, panel A (line 203). Lateral scale bars should be added for all microscopy images. Then, details of the microscopy data acquisition and raw data processing should be also depicted in the respective Material & Methods section.

6) Finally, a brief “Conclusions & Future perspectives” section should be added where the most relevant outcomes found by the authors in this work and the promising future perspectives were outlined. Furthermore, it would be also beneficial to also discuss about the potential future action lines to pursue the topic covered in this research.

Author Response

2

Round 2

Reviewer 1 Report

Comments and Suggestions for Authors

I have no comments